# Exploring organic chemical space for materials discovery using crystal structure prediction-informed evolutionary optimisation

**Jay Johal** [ID] **& Graeme M. Day** [ID] [✉]

Organic molecular crystals offer a broad spectrum of potential applications. The vast number of possible molecules is both an opportunity and a challenge, because of the prohibitive expense of exhaustively searching chemical space to find novel molecules with promising solid-state properties. Computational methods can be applied to direct experimental discovery programmes using high-throughput or guided searches of chemical space. However, to date, such approaches have largely focused on molecular properties, ignoring the often significant effects of the arrangement of molecules in their crystal structure on the molecule's effectiveness for the chosen application. Here, we present an evolutionary algorithm for searching chemical space that incorporates crystal structure prediction into the evaluation of candidate molecules, allowing their fitness to be evaluated based on the predicted materials' properties. As a demonstration, the crystal structure-aware evolutionary algorithm is applied here to a search space of organic molecular semiconductors, demonstrating that the inclusion of crystal structure prediction in the fitness assessment outperforms searches based on molecular properties alone in identifying molecules with high electron mobilities.

Applications of molecular crystals are diverse, including areas such as pharmaceuticals[1], organic electronics[2,3], optical materials[4,5], and porous materials for gas storage and separation[6]. The discovery of functional molecular materials has traditionally relied on trial-and-error experimentation, combined with chemical intuition and the development of empirical rules of crystal engineering. However, computation-led approaches have developed rapidly in recent years, where molecules are assessed in silico to prioritise the most promising candidates for synthesis and characterisation. Since the properties of interest often depend strongly on the arrangement of molecules in their crystal structure, a critical computational method is crystal structure prediction (CSP), which generates and ranks the likely crystal packing possibilities of a molecule by exploring the lattice energy surface for the lowest energy local minima[7,8].

CSP is now commonly applied to pharmaceutical materials for the anticipation of polymorphism[9–12], has guided the discovery of porous molecular crystals[13–16] and has been applied to energetic materials[17,18], photomechanical crystals[19] and organic electronics[20–24]. With effective CSP methods in hand, a key challenge for their application in materials discovery is to efficiently search for promising new molecules for the intended function. The huge number of possible organic molecules[25,26] offers a vast design space for functional molecular materials, but precludes exhaustive searches of this chemical space. Evolutionary algorithms (EAs)[27,28], a class of population-based optimisation techniques inspired by biological evolution, have been demonstrated to efficiently search large chemical spaces for the best candidates. The fitness of all molecules in the population is evaluated, and the candidates that are considered fittest are more likely to be selected as

School of Chemistry and Chemical Engineering, University of Southampton, Southampton, UK. [✉]e-mail: G.M.Day@soton.ac.uk

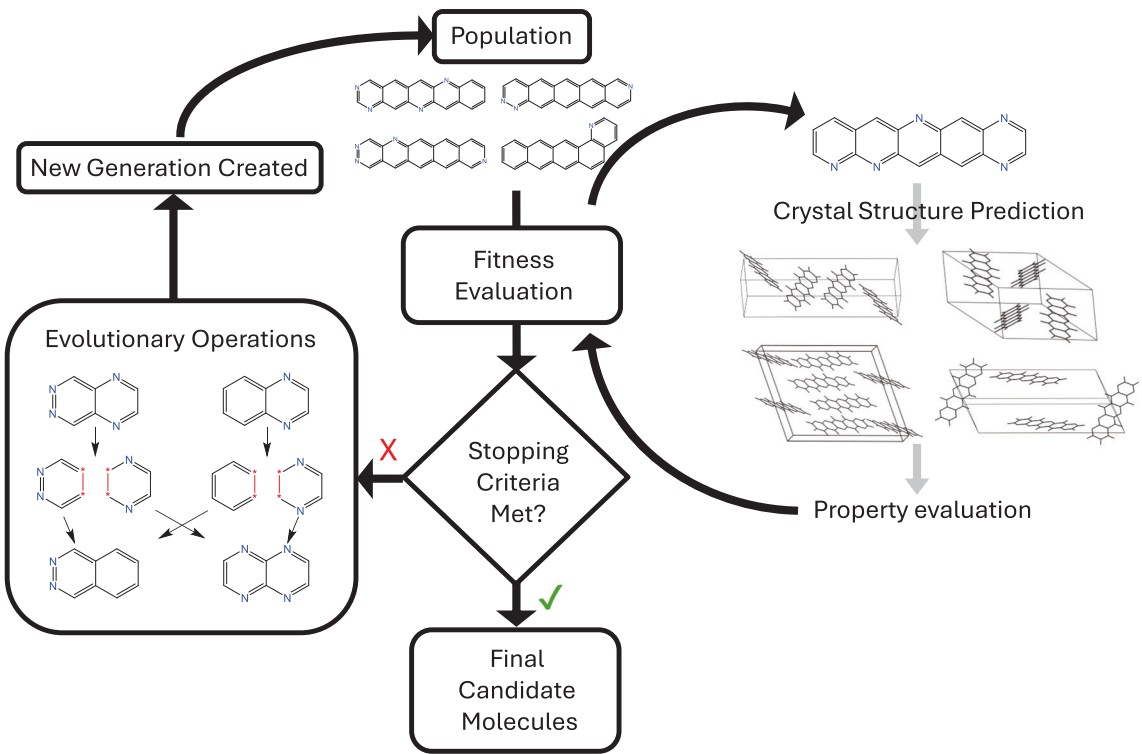

**Fig. 1 | Example CSP-EA workflow.** Schematic of the CSP-informed Evolutionary Algorithm for property optimisation.

parents to generate children, carrying forward characteristics of highly ranked candidate molecules to the next generation of molecules. However, even when only needing to computationally evaluate a small fraction of possible molecules, the computational expense of CSP has meant that chemical space searching methods have thus far been blind to crystal packing, instead relying on the evaluation of molecular properties that are expected to correlate with materials properties[29,30]. This lack of crystal structure information in molecular fitness evaluation has naturally limited the effectiveness of EAs at optimising materials properties that are strongly influenced by crystal packing.

In place of CSP, it is tempting to use a template crystal packing for all molecules, assuming the same, commonly adopted crystal packing motif to predict materials properties of candidate molecules[20,24,31]. However, due to the weak intermolecular forces that govern the stability of molecular crystals, small alterations to a molecule that are made to tune its properties can have a large effect on the preferred crystal packing. Therefore, an assumed packing motif is unlikely to be representative across even small areas of chemical space.

Until recently, the largest reported CSP studies have been limited to 10's of molecules[23,32,33]. However, developments in the efficiency of CSP methods and their deployment on large-scale high–performance computing systems has made it possible to perform high-quality CSP on 100–1000's of molecules on reasonable timescales[34]. This ability to perform CSP at large scale and with excellent reliability in a fully automated manner[34] offers an exciting possibility that we demonstrate in this work: evolutionary exploration of molecular materials where optimisation is directed by the predicted properties of molecules' most likely crystal structures. This is achieved by the incorporation of fast, automated CSP within an EA, allowing the calculation of molecular fitness to be informed by the simulated properties of its landscape of stable predicted crystal structures.

As a demonstration of the approach, we apply the CSP-informed EA (CSP-EA) to small molecule organic semiconductors (OSCs). This class of molecules has applications, such as organic light emitting diodes (OLED), photovoltaic devices (OPV) and field effect transistors (OFET), for which a key property that determines effectiveness is the

charge carrier mobility. OSCs are a challenging class of molecules for chemical space exploration because small modifications to their chemical structures can significantly alter their optical and electronic properties, and charge carrier mobilities are sensitive to small changes in crystal packing[35]. Thus, effective chemical space exploration requires guidance from molecular and materials property predictions. Common modifications that have been explored in developing small molecule organic semiconductors include introducing electron-withdrawing or donating groups, substituting heteroatoms into polyaromatic regions and increasing the length of the conjugated systems.

This work demonstrates that, by embedding CSP within the fitness evaluation and assessing molecules by either the predicted property of the lowest energy predicted crystal structure or a landscape-averaged property, an EA is able to optimise to molecules whose crystal structures have much higher predicted charge carrier mobility than molecules that have been optimised to against a single-molecule property, the reorganisation energy. Furthermore, we demonstrate that the CSP-informed EA can be made computationally efficient by assessing incompletely sampled, but effective CSP landscapes to inform materials property evaluations.

## Results
### Efficient CSP sampling
The structure of CSP-EA is summarised in Fig. 1. The key advance that we demonstrate in this work is to include CSP within the evaluation of each molecule in the evolving population of candidates; this development makes the predicted crystal structures available for the molecular fitness evaluation, which guides the EA's path through chemical space. To achieve this, CSP calculations had to be fully automated from a line notation description of the molecule (i.e., an InChi string) through structure generation, lattice energy minimisation and property assessment. Development of the method also required a careful assessment of the necessary level of CSP to effectively guide an evolutionary search, balancing completeness of the predictions with the cost of the calculations.

**Table 1 | Summary of example CSP sampling schemes investigated, showing the total number of target crystal structures in each scheme, the number of global energy minimum crystal structures from a comprehensive CSP sampling that were recovered in the reduced samplings (for the 20 benchmark molecules), and the percentage of low energy structures recovered (those within 7.2 kJ mol$^{-1}$ of the global minimum for each molecule)**

| Space group | Space group number | Cumulative % Occurrence from the CSD | Sampling Schemes | | | | |
|---|---|---|---|---|---|---|---|
| | | | SG14-500 | Top5-500 | Top10-500 | Top10-2000 | Sampling A |
| $P2_1/c$ | 14 | 39.03 | 500 | 500 | 500 | 2000 | 2000 |
| $P2_12_12_1$ | 19 | 55.99 | - | 500 | 500 | 2000 | 1000 |
| $P\bar{1}$ | 2 | 72.33 | - | 500 | 500 | 2000 | 1000 |
| $P2_1$ | 4 | 81.51 | - | 500 | 500 | 2000 | 1000 |
| $Pbca$ | 61 | 86.48 | - | 500 | 500 | 2000 | 1000 |
| $C2/c$ | 15 | 90.76 | - | - | 500 | 2000 | 1000 |
| $Pna2_1$ | 33 | 92.61 | - | - | 500 | 2000 | 500 |
| $Cc$ | 9 | 93.68 | - | - | 500 | 2000 | 500 |
| $Pca2_1$ | 29 | 94.64 | - | - | 500 | 2000 | 500 |
| $C2$ | 5 | 95.56 | - | - | 500 | 2000 | 500 |
| Total Structures Sampled | | | 500 | 2500 | 5000 | 20,000 | 9000 |
| Number of global minimum structures recovered | | | 12 | 15 | 17 | 19 | 19 |
| Proportion of low energy structures recovered (%) | | | 25.7 | 44.0 | 63.8 | 77.1 | 73.6 |

Space groups are listed in decreasing order of occurrence in the Cambridge Structural Database (CSD)[38]. The cumulative occurrence shows the proportion of Z'=1 organic molecular crystal structures occurring in the given space group and all more frequently observed space groups. The full set of assessed sampling schemes is shown in Supplementary Table 1.

CSP requires a sampling of the structural degrees of freedom defining crystal packing, and evaluation of the relative stabilities of the resulting structures, which correspond to local minima on the lattice energy surface. While large-scale, fully automated CSP has been demonstrated recently[34], performing comprehensive CSP for each sampled molecule during the EA search would be computationally challenging, even leveraging the inherent opportunity for parallel computing as part of a generational EA. As an estimate of scale, in our previous study where an EA was applied to optimise the molecular properties of the azapentacenes[29], each EA run sampled 1544 molecules on average and, due to the stochastic nature of an EA, multiple repeats are required to ensure a complete search. Therefore, we expect that optimisation of charge mobility will require CSP on thousands of molecules and, so, explored cost-effective, coarser CSP sampling schemes for use within the EA.

The sampling scheme's aim is to explore the lattice energy surface to locate the most important crystal structures. In our work, we apply a low-discrepancy, quasi-random sampling of structural degrees of freedom[36]. To be effective, the sampling scheme should capture most low-energy crystal structures of a molecule at a minimum computational cost. We take advantage of two observations in designing these CSP sampling schemes: the uneven occupation of space groups for observed crystal structures of organic molecules[37], and the tendency for CSP to locate the lowest energy structures early in a search[36]. 20 benchmark molecules (Supplementary Fig. 1) are chosen to evaluate the cost and effectiveness of a range of reduced CSP sampling schemes. As a reference for benchmarking, a comprehensive CSP was performed for each molecule, generating and optimising trial crystal structures within the 25 most commonly observed space groups[38] until 10,000 successfully optimised structures were reached for each space group – a total of 250,000 crystal structures per molecule. The subsampling schemes were limited to between 1 and 10 space groups with a maximum of 2000 structures per space group, based on previous convergence testing[36] and reduced to 1000 or 500 structures per space group to evaluate effectiveness vs computational cost (Table 1 and Supplementary Table 1). Many of these schemes sampled the space groups evenly, while some (e.g. Sampling A in Table 1) were designed with a bias towards the more frequently observed space groups, while maintaining sampling of 2000, 1000 or 500 structures

per space group. In total, these sampling schemes ranged from 500 to 20,000 total crystal structures per molecule.

Each tested sampling scheme is evaluated based on whether it locates the global lattice energy minimum from the comprehensive reference CSP search and the proportion of low-energy crystal structures that it recovers. We define low energy as those within 7.2 kJ mol$^{-1}$ of the global lattice energy minimum for each molecule; this value is taken from a large-scale study of polymorph energy differences[39] as the range that captures 95% of energy differences between known polymorph pairs.

We assess the completeness as the proportion of predicted crystal structures across all 20 molecules located with each sampling. Computational cost of each scheme is measured as the mean time taken per molecule on a 40-core node of a high-performance computer (the University of Southampton Iridis5 cluster). As a reference, the comprehensive sampling required an average of 2533 core-hours per molecule. The cost and completeness of the sampling schemes tested are compared in Fig. 2.

The smallest sampling schemes that we evaluate searched for crystal structures in only space group $P2_1/c$; this is justified by the observation that almost 40% of observed crystal structures (with one molecule in the crystallographic asymmetric unit) occur in this space group[38]. These single-space-group searches are very fast (less than 5 core-hours per molecule) compared to the comprehensive sampling and, with only 500 structures, recovered the global lattice energy minima for 12 of the 20 benchmark molecules, increasing to 15 of 20 with 1000 and 2000-structure searches. However, these single space group searches perform less well at reproducing the entire low energy region of the crystal structure landscapes, only recovering between 25.7% (SG14-500) and 33.9% (SG14-2000) of low energy structures.

Increasing the number of space groups included in CSP to 5 and 10 raises the number of benchmark global energy minima that are located, and the fraction of the low energy region that is recovered in the searches (Fig. 2). The most expensive scheme, Top10-2000 – generating 2000 crystal structures in each of ten space groups, has the best performance, recovering on average 77.1% of low energy crystal structures, but at a cost of approximately 169 core-hours per molecule. By comparison, Sampling A, which biases the sampling based on space group frequency, recovers 73.4% of the low energy crystal structures at

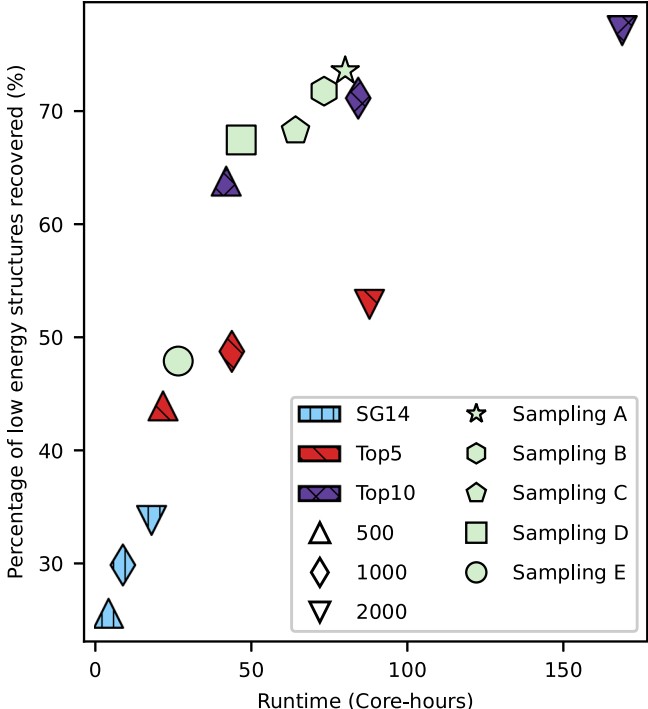

**Fig. 2 | Benchmarking of cost-effective CSP sub-sampling schemes.** Evaluations of CSP sampling schemes against a benchmark set of 20 molecules, comparing the % of low energy structures discovered to more comprehensive sampling. Average time taken per molecule refers to calculations performed on the University of Southampton Iridis5 HPC, with 40 cores per node (2.0 GHz Intel Xeon Gold 6138 processors). Sampling schemes labelled SG14 only generated structures in space group $P2_1/c$, the most commonly observed space group for organic molecules. 'Top5' and 'Top10' include sampling in the 5 and 10 most commonly observed space groups, respectively, each with 500, 1000 or 2000 sampled structures per space group. Samplings A to E involve structure generation in 5 or 10 space groups, with unequal numbers of structures per space group. Source data are provided as a Source Data file.

less than half the computational cost of Top10-2000, and approximately 3% of the computational cost of the reference (comprehensive) CSP. Additionally, Sampling A performs on par with Top10-2000 in recovering the global energy minimum structures; both schemes locate the global energy minimum for 19 out of 20 molecules. The remaining global energy minimum is for tetracyanoethylene, whose global energy minimum structure is found in space group $R\bar{3}$ in the reference CSP, a space group that is not included in any of the sub-sampling schemes; this predicted structure does correspond to the known cubic polymorph of tetracyanoethylene[40], predicted in a sub-group of the full space group, with root mean square deviation (RMSD) in atomic positions of 0.09 Å (evaluated using the COMPACK algorithm[41] with a 30-molecule cluster). Whilst missing this structure highlights the incomplete nature of CSP with these sub-samplings, and the risk that we will occasionally miss important structures in less frequently observed space groups, the results are encouraging in recovering a high fraction of low energy crystal structures with greatly reduced computational cost.

In the following section, we evaluate the effectiveness of these CSP sub-sampling schemes at directing an EA towards molecules with optimised materials properties.

## CSP informed fitness evaluations

We apply the CSP-informed EA method to the optimisation of electron mobility in a search space of azapentacene molecules: all possible molecules consisting of five fused 6-membered rings with any number of nitrogen substitutions. This chemical space includes 135,744 possible molecules.

Polyaromatic compounds, such as azapentacenes, have shown high charge carrier mobilities due to their delocalised π-orbitals[22,29,42–44]. There are four common packing motifs typically adopted by this class of molecules: β, γ, herringbone and sandwich herringbone (Fig. 3). Packings that promote greater orbital overlap result in stronger electronic couplings between neighbouring molecules in the crystal structure and therefore increase the likelihood of such crystals exhibiting higher charge carrier mobilities[22,45]. Aza-substitution of polyaromatic hydrocarbons modifies both the

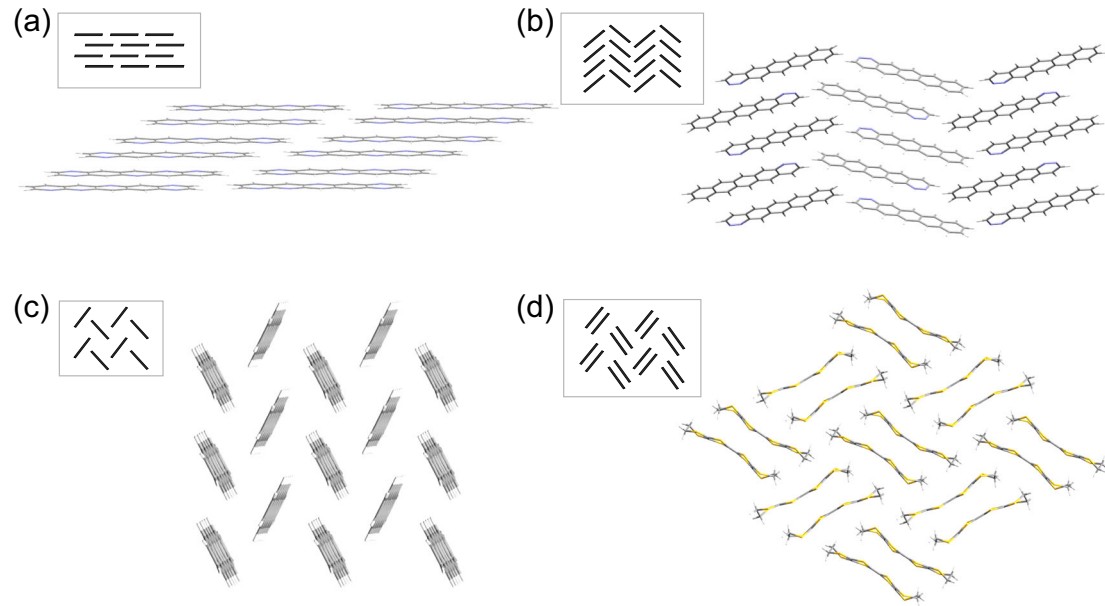

**Fig. 3 | Common polyaromatic packing motifs.** Examples of common packing motifs for polyaromatic molecules. Predicted global energy minimum structures for **a** M1 and **b** M4 exhibiting β and γ packing, respectively. (**M1** and **M4** are molecules generated in the EAs reported in this work), **c** pentacene (polymorph I) showing herringbone packing and **d** the observed crystal structure for molecule BEDT-TTF showing sandwich herringbone packing. Schematics of each packing type are shown in the inset boxes.

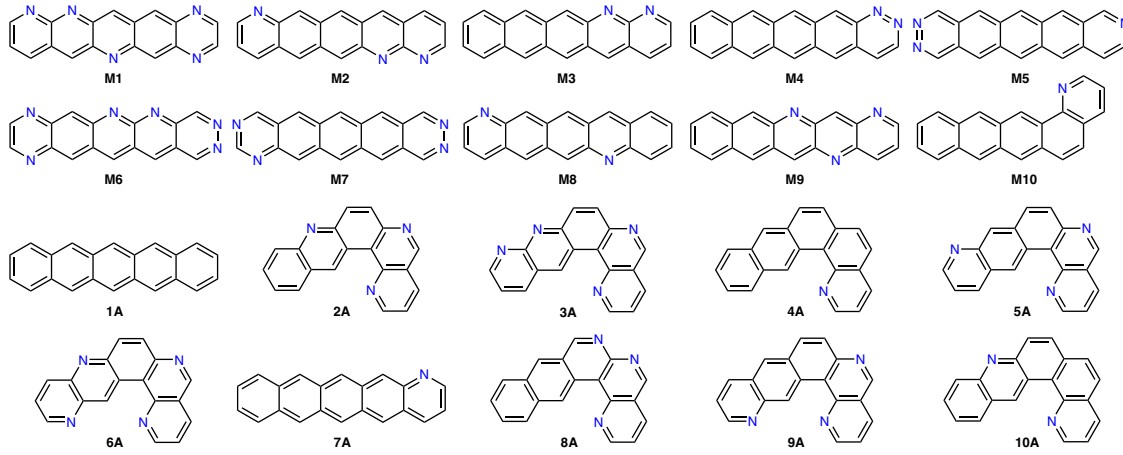

**Fig. 4 | Top identified molecules from the GM_Search and Reorg-EA searches.** Top 10 molecules (**M1-10**) from the CSP-EA global minimum-based searches (GM_Search) and top 10 (**1-10A**) from the previous study minimising $\lambda$ (Reorg-EA).

molecular properties and the potential intermolecular interactions that define the crystal packing, and has been suggested as a route to optimising charge mobility in organic semiconductors[44]. Such molecules do not contain any flexible functional groups or rotatable bonds, simplifying the degrees of freedom required to be sampled by CSP calculations and therefore the computational cost. Thus, we choose this as a moderately-sized chemical search space to demonstrate the effectiveness of CSP-EA, and for comparison to our previous work[29], where we presented evolutionary searches of this same chemical space using molecular properties for the fitness function. Comparison to these earlier results lets us evaluate the benefit of incorporating CSP into the fitness evaluation of the EA.

Starting from random populations of 100 molecules, the evolutionary algorithm is tasked with maximising the electron mobility. Electron mobilities are calculated during the EA using Marcus theory, where electron transport is described by hopping rates, $k$, between molecules in the crystal structure:

$$k = \frac{|V|^2}{\hbar} \sqrt{\frac{\pi}{\lambda k_b T}} \exp\left(\frac{-\lambda}{4 k_b T}\right) \tag{1}$$

in which $\lambda$ is the reorganisation energy (evaluated for electron transport in this work): an energetic cost associated with the change in charge state of a molecule, and $V$ is the electronic coupling between molecules. For efficiency, we apply the analytic overlap model (AOM)[46,47] for estimating electronic couplings from overlap of the relevant molecular orbitals, which has a strong correlation to higher level methods. As reported in the Supplementary Information, Marcus theory calculations using AOM couplings recover the order of observed charge mobilities in crystal structures of different molecules, and the observed relative charge carrier mobilities between polymorphs of a reference set of molecules.

Full details of the evolutionary algorithm, CSP and charge mobility calculations are provided in the Methods section. For each of the tests described below, the EA was repeated three times from independent starting populations. Comprehensive CSP and further electron mobility calculations were performed on the 10 best molecules after aggregating the final populations from these three repeats and we compare to CSP performed on the best 10 molecules from our earlier study[29] (molecules 1A–10A, Fig. 4), where the EA was applied to minimise the reorganisation energy, $\lambda$, a molecular property; we refer to this earlier study as Reorg-EA.

**EA searches using CSP global minima.** We start by assessing the performance of the CSP-informed EA using the properties calculated

from the global energy minimum predicted crystal structure of each sampled molecule: the fitness function to be maximised is the calculated electron mobility of the CSP global minimum.

A main assumption of CSP is that the lowest energy predicted crystal structure is the most likely observable structure. Therefore, for applications where the cost of the fitness evaluation on the crystal is high, the global lattice energy minimum from CSP is considered the most representative single structure to be evaluated[36]. This approach relies on high quality lattice energy evaluations and requires sufficiently complete CSP sampling to provide high confidence that the true global energy minimum has been located. Therefore, CSP subsampling scheme 'Sampling A' was chosen to evaluate each molecule during the EA; this sampling scheme was shown to effectively locate the global energy minima in our benchmark CSP study (Table 1).

The aggregated top 10 for the CSP-EA searches guided by the properties of the global energy minimum (M1–10, top two rows of Fig. 4), referred to as the GM_Search, consists of predominantly linear azapentacene molecules whose global minima, and low energy structures more generally, are dominated by $\beta$ and $\gamma$ packings (Supplementary Note 6). These linear molecule's co-facial packing arrangements, dependent on distance between neighbouring molecules in the crystal, allow for the greatest $\pi$-orbital overlap and therefore increased electronic coupling[22,46,47]. By comparison, with no drive to select molecules with strong intermolecular electronic coupling, the best molecules from Reorg-EA show a mixture of packing types in the low energy regions of their crystal energy landscapes (Supplementary Note 6).

The global energy minimum predicted crystal structures of nine of the ten best molecules display 1D hopping channels. For the remaining molecule, M4, alternating orientation of the cofacially packed columns in the global energy minimum predicted crystal structure (Fig. 3b) gives rise to 2D transport character.

Following the CSP-EA runs, CSP was repeated on the aggregated best 10 molecules with the comprehensive CSP sampling (25 space groups, 250,000 structures in total) and, for comparison, the same calculations were performed for the best ten molecules from Reorg-EA. For the best molecules from the GM_Search, nine out of ten of the global minimum crystal structures found using the CSP sub-sampling Sampling A were also the global minimum from the comprehensive sampling re-evaluation of the molecules, highlighting the effectiveness of the sub-sampling approach to introduce CSP to the EA at reduced computational cost. However, for the remaining molecule, M8, a lower energy structure was found when re-evaluated with more comprehensive CSP, compared to the incomplete sampling performed during the GM_Search CSP-EA. As a result, M8 was ranked considerably lower

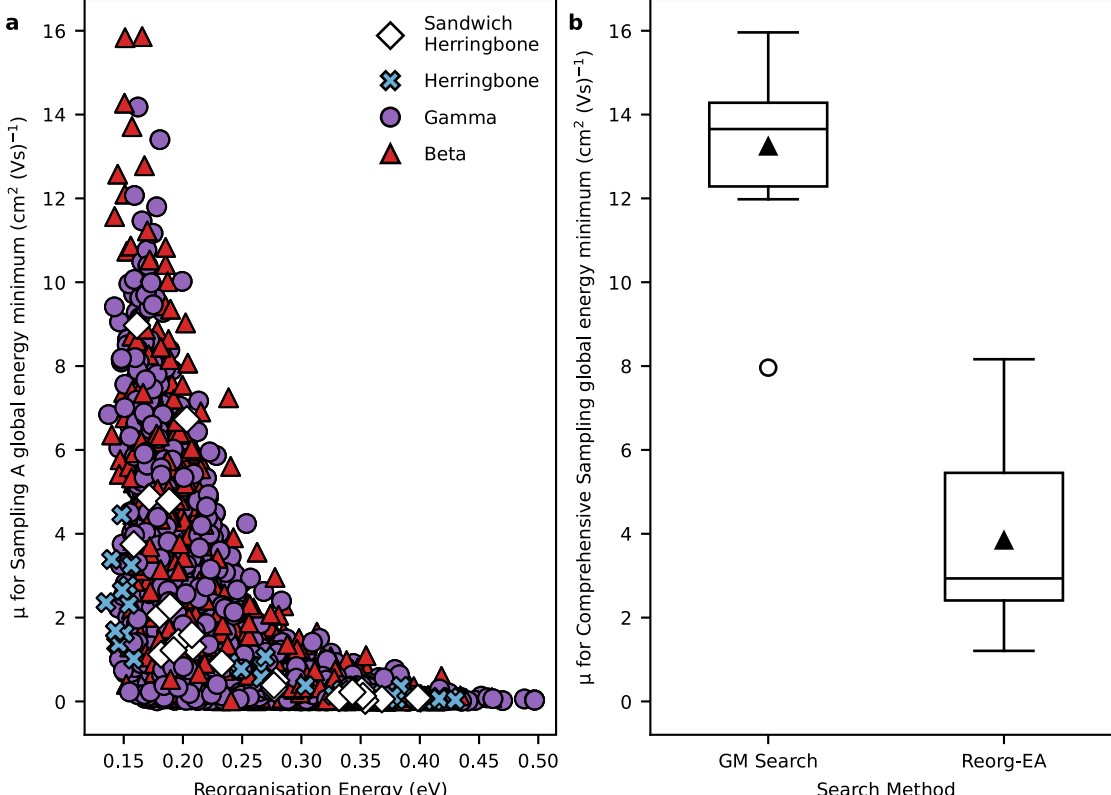

**Fig. 5 | Comparison of the effectiveness of targetting $\lambda$ or $\mu$ of global minimum crystal structures during the search. a** Reorganisation energies and calculated electron mobilities of the global energy minimum predicted crystal structures of the 4061 unique molecules sampled during the GM_Search CSP-informed EA, with data point markers indicating the packing type of the global minimum structure; **b** Boxplots of the electron mobilities calculated for the global energy minimum

predicted crystal structures after comprehensive CSP sampling for the best molecules from the CSP-EA GM_Search (**M1-10**) and Reorg-EA (**1-10A**). The limits of the box indicate the first and third quartiles, the line within the box is the median, the marker in the box is the mean, the whiskers extend to 1.5 times the interquartile range, and markers outside of the whiskers are outliers. Source data are provided as a Source Data file.

upon re-evaluation, due to the more stable crystal packing having a lower calculated mobility (Supplementary Note 6).

Figure 5a shows the calculated electron mobility, $\mu$, for the global energy minimum crystal structure plotted against the reorganisation energy, $\lambda$, for the 4061 unique molecules sampled during the GM_Search. These results illustrate the problem with optimising a molecular property as a proxy for the target material's property. There is a loose relationship between the molecular and materials properties—the highest calculated charge mobilities are recorded for the molecules with the lowest reorganisation energies and high reorganisation energy precludes high charge mobility— which shows why reorganisation energy is commonly chosen as a descriptor for charge mobility[20,29,31,48]. However, there is a wide variation in electron mobility for the molecules with the lowest reorganisation energy. For molecules with low reorganisation energies, high electron mobilities are calculated for those with a preference to pack with $\beta$ or $\gamma$ packing, while those predicted to pack with herringbone or sandwich herringbone motifs generally have low electron mobility (Fig. 5a). Clearly, crystal packing and the variation in the electronic coupling between molecules has a dominant influence in determining the best molecules, so cannot be ignored in methods aimed at optimising $\mu$.

The impact on the outcome of the EA is shown in Fig. 5b, which shows the range of calculated electron mobilities of the global energy minimum predicted structures for the best molecules from the GM_Search and Reorg-EA after re-evaluating their crystal energy landscapes with comprehensive CSP. The CSP-EA delivers molecules whose lowest energy predicted structures have an electron mobility that is, on average, over three times higher than the predicted

crystal structures of molecules that were identified by Reorg-EA, which minimised the molecular reorganisation energy. Even the outlier, M8, for which the lowest energy crystal structure found by sub-sampled CSP was the second lowest energy crystal structure on the comprehensive CSP landscape, has a calculated mobility for the comprehensive CSP global minimum crystal structure that is higher than any of the top molecules from Reorg-EA.

The results in Fig. 5a illustrate why reorganisation energy alone is not a reliable guide for identifying molecules with high charge carrier mobilities in their crystal structures. Furthermore, the assignment of misleading molecular fitness during the EA will be propagated by the selection and crossover operations in the EA, potentially resulting in less favourable molecular motifs being promoted. However, the results suggest an opportunity for future computational cost efficiency improvements by using a multi-fidelity approach to the evaluation of molecular fitness, potentially making CSP unnecessary for molecules for which $\lambda > 0.3$ eV.

**Landscape-averaged EA searches.** The evaluation of molecules based on the properties of the CSP global energy minimum crystal structure has several important issues: it assumes that the CSP sub-sampling scheme is sufficiently complete to have located the true global energy minimum and that the accuracy of the energetic rankings of the crystal structures is good enough to identify the correct structure as the global energy minimum. Furthermore, the approach ignores the ability for organic molecules to exhibit multiple experimentally favourable crystal packings. An alternative approach is to include all low-energy predicted crystal structures in a landscape property average, where the relative energies of predicted structures

**Table 2 | Top 10 molecules for each search as well as the total unique molecules sampled aggregated over the 3 repeats**

| Sampling Scheme | Total unique molecules sampled over 3 repeats | Average Runtime (Core-hours) | Top 10 molecules from sub-sampling search, in rank order |
|---|---|---|---|
| SG14-500 | 3206 | 72,358 | M11, M5, M12, M4, M7, M13, M14, M15, M16, M17 |
| Top5-500 | 3010 | 104,507 | M12, M2, M5, M18, M4, M7, M19, M1, M8, M20 |
| Top10-500 | 3108 | 174,359 | M2, M12, M5, M18, M4, M7, M19, M1, M8, M14 |
| Sampling A | 3089 | 290,601 | M2, M12, M5, M18, M4, M7, M8, M21, M20, M14 |
| Over all searches | 9008 | - | - |
| GM Search | 4061 | - | M1-10 |

Average runtime for each search in core-hours, performed using the UK national HPC Archer2, with 128 cores per node (dual AMD EPYC$^{TM}$ 7742 64-core processors running at 2.0 GHz).

are interpreted as assigning relative probabilities that each low-energy crystal structure would be observed.

Further CSP-EA optimisations were run with the target of maximising a crystal landscape average electron mobility, as defined in Equation (2).

$$\langle \mu \rangle = \sum_{i=1}^{N} \bar{\mu}_i \times \frac{f(E_i) \times exp^{\frac{(-\Delta E_i)}{\tau}}}{\sum_{j=1}^{N} f(E_j) \times exp^{\frac{(-\Delta E_j)}{\tau}}} \quad (2)$$

where $\bar{\mu}_i$ is the isotropic average for the mobility of crystal structure $i$, representing the average of the calculated mobilities along the different potential mobility channels. The sum is then over all low energy crystal structures, i.e. all crystal structures within 7.2 kJ mol$^{-1}$ of the global lattice energy minimum. The fraction on the right of Equation (2) represents the probability that structure $i$ will be observed. The probability takes the form of a Boltzmann weighting[22,49], with the parameter $\tau$ fitted to results from a previous study of lattice energy differences between polymorphs ($\tau = 2.7$ kJ mol$^{-1}$)[39]. The function f(E$_i$) relates to the density of states of predicted crystal structures at energy $E_i$ and is discussed in the Supplementary Note 3.

$\langle \mu \rangle$ can be interpreted as a measure of the propensity of a given molecule to adopt a crystal structure with high electron mobility. Adopting this fitness metric helps by allowing the consideration of multiple potential crystal structures, reducing issues with assigning the molecular fitness during the EA based on the properties of a potentially incorrectly ranked or undersampled global minimum structure. Additionally, $\langle \mu \rangle$ takes into account the existence of potential polymorphs and their effect on final mobilities, allowing greater confidence in the qualitative rankings between sampled molecules. In short, molecules whose low-energy crystal structures consistently exhibit higher mobility will be ranked higher than molecules whose low-energy crystal structures show a mixture of high and low predicted electron mobilities.

To determine the most efficient way to incorporate CSP into the landscape-averaged EA search, CSP-EA was run with four CSP samplings (Table 2), chosen from the Pareto front of Fig. 2, i.e., those with the best balance of computational cost and completeness of the CSP landscape. These samplings range from SG14-500, which is a very fast, minimal CSP by sampling a single space group with 500 trial crystal structures, up to Sampling A, which considers 10 space groups and a total of 9000 trial structures.

The best molecules from these CSP-EA searches are shown in Supplementary Fig. 9 and listed in Table 2. The results of searches with all four levels of CSP sampling reveal a similar favouring as seen in the GM_Search for the linear azapentacenes. In fact, each of the sub-samplings also ranked a varying number of the GM_Search top ten in their own respective top ten counts, highlighting those particular molecules' consistency at favourably adopting high mobility low energy crystal structures. However, for each of the GM_Search top ten molecules (M1-10), the calculated landscape average mobility is lower than the previously calculated global minimum structure's mobility,

indicating that not all the crystal structures sampled showed as high or a higher degree of potential. In particular, landscape averaging of the electron mobility prediction significantly lowers the calculated fitness of molecules M3, M6, M9 and M10 due to the number of low-energy crystal structures on their landscapes that exhibit poor charge mobility. As such, those four molecules were not ranked among the top ten of any of the sub-samplings tested, highlighting how the landscape property average favours molecules with consistently favourable predicted properties among their predicted crystal structures.

Good agreement is found in the best molecules identified with the different CSP sub-samplings, with only 17 unique molecules from the combined top ten rankings from the four sampling settings. This repeated discovery of the top ranked molecules, independently of the level of CSP sampling, is encouraging, and we note that none of the top molecules from the Reorg-EA, which minimised the molecular reorganisation energy, ranked within the top ten for any of the CSP-EA runs. Furthermore, the efficiency of the EA is clear from the small number of molecules that were assessed in each run. Whilst repeatedly finding the same best molecules, the three aggregated repeats of CSP-EA with each sampling scheme sampled just over 3000 unique molecules, which is approximately 2% of the total chemical space. Thus, the EA represents a 50-fold saving compared to high-throughput evaluation of all possible molecules. The average electron mobility across the population increases steeply over the first 10–15 generations before levelling off (Supplementary Figs. 10–15), indicating that search has converged on the best molecules. The three repeats for each level of CSP sampling are quite consistent, with good overlap in the final best ten molecules (Supplementary Figs. 11–15 and Supplementary Table 4); this consistency between repeats is better for searches using landscape-averaged properties than those based on the global minimum energy structure.

Figure 6a compares the landscape-averaged electron mobilities for the best molecules from each of the EAs: Reorg-EA, GM_Search and the landscape-averaged CSP-EA with four CSP sampling schemes. All five variations on CSP-EA show a large improvement over Reorg-EA, with the best molecules showing predicted mobilities approximately three times higher, which re-emphasises the benefit of providing the EA fitness evaluation with predicted crystal structures of the sampled molecules. Figure 6b shows that Reorg-EA located molecules with lower reorganisation energies than the CSP-EA method developed in this work, but these molecules have lower calculated electron mobilities than almost all of the molecules from CSP-EA.

It is particularly encouraging that CSP-EA using the computationally least expensive SG14-500 sampling returns a set of top ten molecules whose landscape averaged electron mobilities, after comprehensive CSP, are only slightly lower than the top molecules returned from CSP-EA using the more complete Sampling A scheme (Fig. 6a). In spite of the notably weaker correlation of SG14-500 evaluations to those from more comprehensive CSP (Fig. 6c) compared to the strong correlation that Sampling A CSP evaluations show (Fig. 6d), this result demonstrates that even an incomplete CSP landscape contains sufficient information on molecules' packing

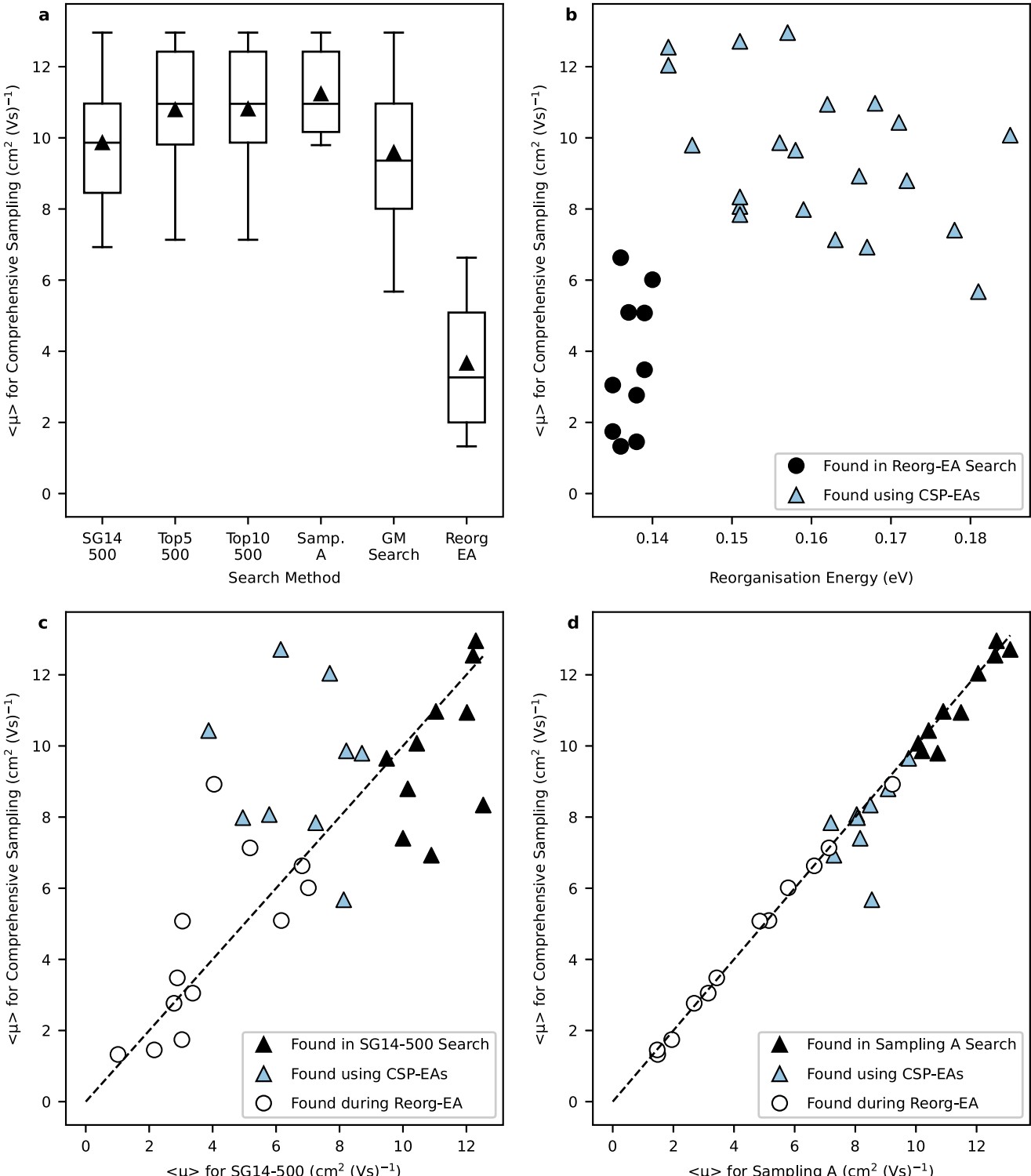

**Fig. 6 | Comparison of the effectiveness of targeting ⟨μ⟩ with CSP-EAs at the different sub-samplings compared to λ for Reorg-EA. a** Comparison of landscape averaged electron mobilities calculated after comprehensive CSP for the top 10 molecules from CSP-EA runs with four different CSP sampling schemes, and comparison to the best molecules from optimisation of the molecular reorganisation energy (Reorg-EA). The boxplot limits indicate the first and third quartiles, with the median and mean represented by a line and a marker, respectively. The whiskers extend 1.5 times the interquartile range. **b** Lack of correlation between landscape averaged electron mobilities for the best molecules from each EA with their reorganisation energy. **c, d** Comparison of landscape averaged electron mobilities from sub-sampling schemes SG14-500 (**c**) and Sampling A (**d**) with the averaged mobility calculated after comprehensive CSP. Black filled data points in parts (**b**–**d**) highlight the molecules optimised using the EA highlighted in that sub-figure: the Reorg-EA molecules in (**b**), those using sub-sampling SG14-500 for CSP-EA in (**c**) and the molecules from CSP-EA with Sampling A in (**d**). See Supplementary Table 3 for more details. Source data are provided as a Source Data file.

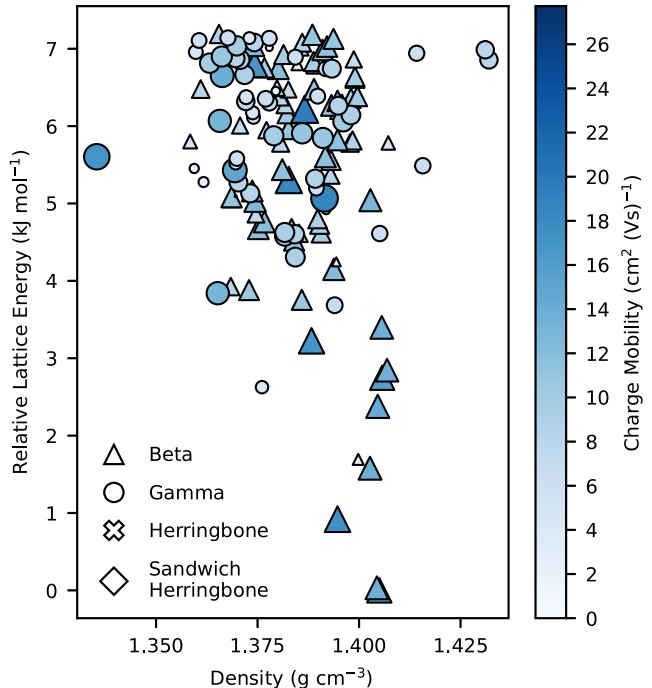

**Fig. 7 | Predicted charge mobility energy-structure-function map for molecule M2.** Predicted energy-density-function map for charge mobilities of each crystal structure for molecule **M2** in the 7.2 kJ mol⁻¹ low energy region. Each data point represents a sampled crystal structure, with the darker and larger the data point the greater the calculated mobilities. Each structure is also classified by its packing motif.

preferences to guide an EA towards high performing molecules. In fact, five of the top ten molecules (M4, M5, M7, M12 and M14) from CSP-EA searches using the most complete CSP sampling, Sampling A, were located in the top ten from the much cheaper CSP-EA runs that used SG14-500 sampling. However, other top ten molecules discovered from the Sampling A CSP-EAs are ranked significantly worse when evaluated at the SG14-500 level of sampling, with M2 and M21 particularly ranked poorly compared to their Sampling A and comprehensive CSP landscape averages. These discrepancies are due to incomplete CSP sampling with SG14-500, which risks missing low-energy crystal structures with good properties. As an example, for M2, which is the top-ranked molecule from the Sampling A CSP-EAs, the global energy minimum ranked crystal structure from SG14-500 CSP sampling, which adopts a γ packing structure, is ranked 7th on the comprehensive CSP landscape, where the low energy structures predominantly adopt β packing (Fig. 7).

The impact of the structures missed by the SG14-500 sampling scheme is shown in Fig. 6c, which compares landscape-averaged electron mobilities from SG14-500 CSP sampling and after the comprehensive CSP sampling; the differences occur due to some of the lowest energy worse performing crystal structures not being sampled by the less complete search. The correlation, albeit weak, between landscape-averaged electron mobilities from SG14-500 and the comprehensive CSP, as well as its reduced algorithmic cost as part of an EA, highlights its potential for multi-fidelity or coarse initial searching of chemical space at reduced computational expense.

As the CSP sampling used during the CSP-EA runs is increased, so does the average electron mobility when assessed with comprehensive CSP (Fig. 6a), due to increasing correlation of the landscape-averaged properties from the sub-sampled CSP with the comprehensive CSP landscape averages (Supplementary Table 3). Indeed, Top5-500 and Top10-500 each share 8 of the same top ten molecules as Sampling A, with the same identified top 6 molecules as well. Top10-500

outperforms Top5-500 by ranking the first and second best sampled molecules in the same order as Sampling A. However, it comes at an increased cost at 60.0% of the cost of Sampling A compared to 36.0% for Top5-500.

Additionally, the tighter range of average values from the landscape-average Sampling A CSP-EA searches compared to the GM_Search CSP-EA, even though they have the same CSP sub-sampling scheme, further showcases the effectiveness of the landscape averaging at optimising to molecules whose crystal structure landscapes have a larger fraction of high mobility, low energy crystal structures.

## Discussion

We have presented the development of an evolutionary algorithm, CSP-EA, for exploring chemical space to inform the discovery of new materials. The key innovation in this work is to embed CSP within the evolutionary algorithm, which allows the predicted properties of low-energy crystal structures to guide the optimisation process towards the most promising molecules. This is made possible by fully automating the CSP process from InChi string notation of the candidate molecule to the final crystal structure landscape, and is made computationally tractable by assessing low-cost CSP sampling schemes for recovering the majority of low-energy crystal structures available to a molecule.

The method, CSP-EA, is designed to optimise towards molecules whose most likely observable crystals have optimised properties. CSP-EA is demonstrated on the chemical space of azapentacenes as n-type organic semiconductors, where we aim to maximise the electron mobility. The results demonstrate that even a fast and incomplete sampling of the crystal structure landscape provides sufficient information to the EA to guide optimisation towards molecules with superior predicted materials properties (ca. 3 × higher electron mobilities) than an EA using molecular properties alone.

Additionally, we have shown the benefits of using a landscape averaging approach to evaluate the likely materials properties associated with a molecule, which produces molecules with the best likelihood to form high charge mobility crystal structures. We also find that EAs based on the predicted properties of only the global energy minimum crystal structure perform well, which will be an effective strategy for situations where the cost of the property prediction is too high to be applied to all low-energy crystal structures.

Beyond the demonstration on azapentacenes, the CSP-EA should be applicable to molecules with stronger, more directional intermolecular interactions, such as hydrogen bonds; the same CSP methods have been applied successfully to such molecules[14,15,50]. Further challenges to this method involve application to chemical spaces with increased molecular flexibility and to multicomponent crystals such as cocrystals and salts, where the increased structural dimensions will impact the cost of the EA search, largely due to the increased cost of CSP, which would require greater sampling to generate representative crystal structure landscapes. However, the inherently parallel nature of the generational EA used here, as well as the CSP procedure, allows effective scaling with improving access to computational resources. Furthermore, the identification of well-performing and efficient sub-sampling schemes provides an opportunity in the future to leverage funnelling or multi-fidelity approaches. Such implementations could allow either larger or more complicated chemical spaces to be investigated, including those for which the fitness evaluation has no molecular property analogues, such as searches for extrinsically porous molecular crystals[15,16].

We view these developments as being potentially transformative in computer-guided materials design and, furthermore, provide a framework for general application of CSP-informed fitness evaluations for generative models for functional materials.

## Methods

### Evolutionary algorithm

Using RDKit[51] molecular objects, an initial population of molecules was generated by addition of fragments until a molecular size criterion is met. Fragment addition is performed using special characters (*) to denote attachment points inputted as SMILES strings e.g., c1c**cc1 as a benzene ring attachable fragment. Fusing fragments to form phenalene or pyrene-like molecules was not permitted. To each generated molecule, a specified number of random mutations are then performed to further increase variety of starting populations using SMARTS strings to detect mutable positions and randomly replace e.g. [[#6R1&H], [#7R1&H0]] allowing the replacement of a carbon in a ring attached to a single hydrogen with a nitrogen in a ring not attached to a hydrogen, as well as vice versa. Additional possible mutation involves randomly fragmenting the initial molecule and recombining the fragments to create an isomer of the original.

Once the initial population is generated, each molecule undergoes fitness assessment which, for this study, includes molecular geometry optimisation, reorganisation energy calculation, CSP, and Marcus theory hopping rate calculations on the generated crystal structures before returning a final fitness value for the molecule.

If the required number of generations has not been met, then the top evaluated molecules from the population are copied unchanged to the next population to be evaluated, up to the specified number of elite molecules. Then, to build the rest of the next population to the specified population size the current population undergoes binary tournament selection to choose parent molecules to undergo crossover operations to build the next generation. Binary tournament selection involves randomly selecting two sets of two potential parent molecules. The fitness values of the parents are then compared, and the higher performing parent is selected 75% of the time to return two parents. The worse performing parent is chosen 25% of the time to aid greater exploration of the chemical space. For the two remaining parents ($parent_i$ and $parent_j$), potential fragmentation points are determined and parents are randomly split into an A and B fragment. Fragment $A_i$ is combined with fragment $B_j$, and $A_j$ with $B_i$ to create two new child molecules. If the child molecules fit the set molecular size criteria, they have a probability to undergo a mutation and recombination operation before being added to the next population. The next population then becomes the current population and undergoes fitness assessment, repeating the procedure until the required total number of generations has been reached. For more detail, please see ref. 29.

The parameters controlling elitism, cross-over and mutation rates have been kept the same as in ref. 29 in order to compare the effect of the fitness assessment (Table 3).

### Crystal structure prediction

For each molecule sampled in an evolutionary algorithm search, a InChi string was generated along with an InChiKey for unique file naming. Using RDKit[51] an initial 3D geometry was generated before undergoing an initial optimisation using UFF[52]. This geometry was further optimised by density functional theory (DFT) using Psi4 (v1.7)

with the B3LYP functional and 6-311+G(d,p)[53,54]. Apart from the inclusion of diffuse functions, this functional and basis set is the same as used in our recent validation of the CSP workflow on over 1000 molecules[34]. CSP was performed using a quasi-random sampling method by mapping elements of a Sobol vector to the crystal structures degrees of freedom (molecular positions, orientations and unit cell angles and lengths) constrained by the space group the trial structure was generated within. The DFT-optimised molecular geometry was held rigid during the crystal structure generation and lattice energy minimisation steps. Initial crystal structures were checked for clashes between molecules and relieved by expansion of the unit cell[36].

All accepted structures are minimised using a 3 step approach: initially using the PMIN package with atomic partial charges for the electrostatic model, followed by re-optimisation using DMACRYS (v2.3.1.1)[55] with atomic partial charges and a pressure of 0.1 Pa before a final lattice energy minimisation with no applied pressure using a more detailed electrostatic interactions model consisting of atomic multipoles, using DMACRYS. The force field used for each stage was the FIT empirically parametrised exp-6 repulsion-dispersion model[56] with electrostatics derived from the B3LYP/6-311+G(d,p) charge density. Atomic multipoles, up to hexadecapole for all atoms, were calculated from a distributed multipole analysis, performed using GDMA (v2.3.3)[57]. Atomic partial charges were fitted to the molecular electrostatic potential derived from the atomic multipole model, using the MulFIT (v2.1.03) programme[58].

For each chosen CSP sampling scheme, crystal structures were generated with one molecule in the asymmetric unit ($Z' = 1$) until the required number of successfully energy minimised crystal structures had been generated in each space group, as defined by the sampling scheme. Reference calculations for benchmarking of sampling schemes were performed in $P2_1/c$, $P2_12_12_1$, $P\bar{1}$, $Pbca$, $P2_1$, $C2/c$, $Pna2_1$, $Cc$, $Pca2_1$, $C2$, $P1$, $Pbcn$, $Pc$, $P2_12_12$, $P4_32_12$, $P4_1$, $P3_2$, $Fdd2$, $Pccn$, $P2/c$, $P6_1$, $I4_1/a$, $C222_1$, $P4_2/n$ and $R\bar{3}$. Duplicate crystal structures were then removed from the final structure set by comparing predicted powder X-ray diffraction patterns (PXRD), generated using PLATON[59], of all structures across the sampled space groups for each molecule that are within 0.05 g cm$^{-3}$ in density and within 1 kJ mol$^{-1}$ in energy. PXRD patterns were compared using constrained dynamic time-warping. After duplicate removal on the congregated final set of crystal structures, either the global minimum or all the structures within 7.2 kJ mol$^{-1}$ of the global minimum were taken to evaluate charge mobilities.

The crystal structures sampled in the low energy window from each sub-sampling were structurally compared to the structures in the comprehensive sampling by comparing using the COMPACK algorithm[41], comparing 30-molecule clusters from reference and comparison crystal structures.

### Marcus theory charge mobility

All mobility calculations as part of the evolutionary algorithm were performed using Marcus theory calculations at $T = 300K$.

Hopping rates were determined between all molecule pairs within a 30 Å cut-off of a central unit cell with at least one atom-atom distance shorter than the sum of their van der Waals radii plus 1.5 Å. From these rates, we generate a $3 \times 3$ diffusion matrix (**D**). A mobility matrix was then determined by applying the Einstein relation before taking the eigenvalues to determine the mobility for each Cartesian direction, with $Z$ defined as the highest mobility plane. A supercell was used for this dimer detection, expanded to contain all relevant dimers.

$$\mu = \frac{\mathbf{D}}{k_b T} \tag{3}$$

Electronic coupling values were calculated using the analytic overlap method using the pyAOMlite library which has a far lower associated computationally cost when compared to typical electronic

**Table 3 | Parameters used for the evolutionary algorithm**

| Parameter | Parameter Choice |
| --- | --- |
| Population Size | 100 |
| Initial number of mutations | 500 |
| Elite Molecules count | 10 |
| Mutation Chance | 5% |
| Recombination Chance | 5% |
| Total number of generations | 30 |

coupling ab initio or DFT methods, that has been fitted to reproduce electronic couplings between $\pi$-conjugated organic molecules. For a full description see ref. 47. In short, the frontier molecular orbitals (FMOs) of the isolated molecules in the dimer are projected from a DFT calculated atom-centred gaussian-type orbital using the CP2K package[60] (v8.2) into a Slater-type orbital basis with only the s and p type angular momentum contributions. The description is further simplified as the projected p-orbitals parallel to the plane of $\pi$-conjugation in the molecule are neglected along with the s orbitals, only considering p-orbitals orthogonal to the plane of $\pi$-conjugation corresponding to p-orbitals typically involved in $\pi$-bonding only. This minimalistic representation of the FMOs is then mapped onto the molecules in the dimer and used to calculate the overlap integral $S_{ab}$[46,47]. In this study, the FMO of the donor and the acceptor are the same as they are the same molecule and have the same geometry. Therefore, only one DFT single-point calculation, at the PBE/DZVP-GTH level of theory, was required for each sampled molecule as part of the fitness evaluation procedure. The electronic coupling $V_{AB}$ was then calculated assuming a linear relationship between the electronic coupling and the transfer integral. With a universal scaling constant, $C$, determined to equal 9463 meV when correlated against scaled projector operator-based diabatization (sPOD) calculations[61].

$$|V_{AB}| = C|S_{ab}| \tag{4}$$

Reorganisation energies were calculated using the four-point approach[29,62-64]

$$\lambda = [E_n(R_0) - E_0(R_0)] + [E_0(R_n) - E_n(R_n)] \tag{5}$$

where the energies and optimised geometries were calculated using Psi4 (v1.7)[54] with the B3LYP functional and 6-311+G(d,p)[53]. As such the external reorganisation energy is excluded, as its contribution has been found to be negligible[65].

## Data availability
The CSP structures and their predicted properties for each top 10 molecule, and associated mobilities at varying levels of sampling, are available at https://doi.org/10.5258/SOTON/D3613[66]. Additionally available are CSP structures and calculated mobilities for each sampled molecule from the EA runs. Source Data are provided with this paper. Additional data can be found in the Supplementary Information.

## Code availability
The code for the CSP-EA procedure is available at https://gitlab.com/mol-cspy/. The exact version used in this work can be found in ref. 67.

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

## Acknowledgements

This project has received funding from the European Research Council under the European Union's Horizon 2020 research and innovation programme (grant agreement no. 856405 (J.J. and G.M.D)). The authors acknowledge the use of the IRIDIS High Performance Computing Facility, and associated support services at the University of Southampton, in the completion of this work. Via our membership of the UK's HEC Materials Chemistry Consortium, which is funded by the EPSRC (EP/R029431), this work used the ARCHER2 UK National Supercomputing Service (https://www.archer2.ac.uk). We are grateful to C.Y. Cheng and J.E. Campbell for the initial development of the evolutionary algorithm the CSP-EA method builds upon.

## Author contributions

J.J. developed the software required for the CSP-EA method, performed the calculations, curated the data, analysed the results and prepared the manuscript. G.M.D. conceptualised the study, supervised the work performed, discussed the results, managed funding acquisition and performed review and editing of the manuscript.

## Competing interests

The authors declare no competing interests.
