## [Transparent Peer Review file · Nature Communications]

Exploring organic chemical space for materials discovery using crystal structure prediction-informed evolutionary optimisation

Corresponding Author: Professor Graeme Day

Version 0:

Reviewer comments:

Reviewer #1

(Remarks to the Author)

Johal and Day combine crystal structure prediction (CSP) and evolutionary algorithms (EA) to explore organic molecular chemical space for materials discovery. The proposed CSP-EA framework is demonstrated on azapentacene derivatives with the objective of optimizing charge carrier mobility, and comparisons are made against molecular-property-only optimization (Reorg-EA). Their results show that the approach incorporating CSP in fitness evaluation outperforms searches based solely on molecular properties in identifying materials with high electron mobility. However, several concerns limit the impact and clarity of the current submission.

Major comments:

1. In Table 1, the choice of the CSP subsampling scheme is fully described, but the reason for the choice of the number of structures is not clearly described. Is it based on empirical design choices, or has convergence testing or benchmarking been performed?
2. The authors claim that even the cheapest SG14-500 can yield molecules with competitive predicted mobilities. But the paper only states: "Four of the top 10 molecules....", without a quantitative result for the full picture. It would be interesting if the authors could show the RMSE or bias between the SG14-500 and the comprehensive CSP predicted mobility, or the fitness correlation scatter plot between SG14 and the comprehensive CSP.
3. The CSP-EA frameworks show good results when applied to azapentacenes. However, it is a rigid and planar structure. What about the performance for soft, flexible structures? Or salts or molecules with H-bond interactions?
4. The authors mentioned the EA is run three times per setting, and the results are averaged. However, there is no quantification of variance across runs in terms of best fitness, diversity of the results (top 10 molecules), or convergence behaviors. It will be more informative if the authors could add standard deviation, the overlap of the top 10 molecules in three runs, and the single converged trajectory for each time for Figure S10.
5. It would be more informative if the authors could discuss how often each molecule was selected or how stable the rankings for the top 10 molecules are. It can support the stability and reproducibility of this approach.
6. The authors used B3LYP functional and 6-311+G for optimization. Is there any benchmark for it?

Minor comments:

1. In Figure 1, the molecular structures are not clear and are not consistent.
2. Overall, the quality of the figures and Tables is not satisfactory.
2. The format of references is not consistent.
3. Typos throughout the text.

(Remarks on code availability)

Reviewer #2

(Remarks to the Author)

(Remarks on code availability)

They have made a very detailed introduction website (<https://mol-cspy.readthedocs.io/en/latest/>), which has very detailed instructions and examples. And the results are almost the same as expected

Reviewer #3

(Remarks to the Author)

The manuscript by Johal and Day presents an evolutionary search workflow for the discovery of functional organic molecules, based on fitness measures that are obtained from the relevant crystalline forms of the candidate molecules through an implementation of crystal structure prediction algorithms.

The article uses the case of azapentacenes as a demonstration of the proposed methodology, where the charge mobility is used as the target property to be evaluated for either the global minimum of the accompanying CSP, or for a pool of low-lying minima of the energy landscape.

Overall, the manuscript is well-written and the presentation is successful in delivering the idea, showcasing its performance, and indicating the challenges and possible areas for further work.

Various interesting discussions and insights are offered, e.g., about the correlation of molecular and materials properties as optimization target and the following suggestion about using a multi-fidelity approach; and the idea of landscape-averaged properties for the EA are in particular interesting, especially for its relevance to the cases where "the fitness evaluation has no molecular property analogues".

There are, however, a few points about the presentation of the materials that can hinder the usefulness of the work. As a result, and given the above-mentioned quality of the work, I recommend the publication of the article in nature communications with the below-mentioned points addressed.

(1) A general point about the figures:

- Fig2: Distinguishing between dark points (Top5 and Top10) in b/w figure is very hard; and larger font sizes for the in-figure texts (labels and legends) would be very helpful.
- Figs 4, 5, and 6: The font sizes for texts are fairly small.
- Generally, why all b/w plots in online article?

(2) Figure 2 (and earlier mention in line 140) reference the node-hours for a 40-core node. Later, (e.g., Table 2), the results are presented from runs on 128-core nodes. Although I understand that the cores' performance and parallelism efficiency might be different, but it would be better to unify the presentation (e.g., in "core-hours") for clarity.

(3) Line 302: the " average mobility for crystal structure i": is this average over the same set for which the summation is performed?

(4) A few typos:

- Line 238: "the best molecules molecules from Reorg-EA show a mixture"
- Line 296: "A an alternative approach is include ..."
- Line 303: Un-paired open parenthesis

(5) The labeling of global minima:

In the text there are references to the true global minimum. As a general point, I would argue that this might not be the best approach. As a particular example, in line 278, I found the

"... the subsampled CSP found the incorrect global minimum energy crystal structure, ..."

a bit unassuming and misleading at the same time! In this case, the one that was found in CSP-EA was not "incorrect"; rather, the comprehensive CSP found a better minimum. In turn, the one that was found in the comprehensive CSP is not necessarily "correct"; as another search might land to an even better structure.

I think this classification of the global minima into correct/incorrect or true/untrue is not the best presentation.

(Remarks on code availability)

I have overviewed the code and its documentation. Although I didn't run the code in the sense of reproducing the manuscript's results; but I believe the source code is well-organized and, especially, a comprehensive online documentation is provided with details about utilizing the code and nice examples of the usage of various implemented feature (e.g., structure generation, analysis of landscape, etc).

Version 1:

Reviewer comments:

Reviewer #1

(Remarks to the Author)

The authors have appropriately addressed my comments/suggestions; I'm happy to recommend publication in Nat Commun.

(Remarks on code availability)

Reviewer #2

(Remarks to the Author)

(Remarks on code availability)

The code is well organized and the test results are close to the data they provide.

Reviewer #3

(Remarks to the Author)

The points mentioned in the original review are properly addressed in the revised manuscript; hence, I can recommend it for publication in the nature communications.

I just have a minor comment about the revised version: while the figures are improved significantly, I found the Figure 3 (especially sub-fig [3a]) to be replaced with a new one which is somewhat less clear compared to the original figure. I'm not sure why this change was made, however, I think the first figure (as it was) was better in showing the details.

(Remarks on code availability)

We thank the reviewers for their helpful comments on the manuscript. Please find our point-by-point response to reviewer comments below; reviewer comments are shown in red, followed by our response in black. An annotated version of the manuscript and supporting information file is provided with revisions highlighted in red.

Reviewer #1 (Remarks to the Author):

Johal and Day combine crystal structure prediction (CSP) and evolutionary algorithms (EA) to explore organic molecular chemical space for materials discovery. The proposed CSP-EA framework is demonstrated on azapentacene derivatives with the objective of optimizing charge carrier mobility, and comparisons are made against molecular-property-only optimization (Reorg-EA). Their results show that the approach incorporating CSP in fitness evaluation outperforms searches based solely on molecular properties in identifying materials with high electron mobility. However, several concerns limit the impact and clarity of the current submission.

Major comments:

1. In Table 1, the choice of the CSP subsampling scheme is fully described, but the reason for the choice of the number of structures is not clearly described. Is it based on empirical design choices, or has convergence testing or benchmarking been performed?

Thank you for your comment. From previous experience and testing of our CSP approach we find that the lowest energy crystal structures on a CSP landscape are typically located early and frequently during a quasi-random search. Typically, it is the complete location of higher energy crystal structures that require more extensive sampling.[1,2] From the results in Ref. [1], which examined the convergence of the quasi-random structure generation process, it can be seen that for the trial molecules examined in the study that the number of unique structures found in the lowest 15 kJ mol⁻¹ region of the landscape begins to plateau after 2000 successful minimisations. As such, this motivated the choice of 2000 structures as the largest sampling size for an individual space group as can be seen in Supplementary Table 1. This choice was further supported from the benchmarking for the Top10-2000 CSP sampling which can be seen to identify 19 out of the 20 same lowest energy minima as the comprehensive search and 77.1 % of the low energy, 7.2 kJ mol⁻¹, window. The motivation behind the use of 500 structures and 1000 structures per space group then simply followed as factors of ¼ and ½ to investigate their effectiveness vs the predictable reduction in computational cost.

While the Top5-N and Top10-N subsampling schemes were based on evenly sampling each space group, we also included five subsampling schemes that also factored in the relative importance and known differences in converging the search in different space groups. To factor in space group importance, we consider the frequencies of occurrence of each space group in the Cambridge Structural Database (CSD). However, these frequencies drop off quickly, from $P2_1/c$ at ~39 %, to the tenth most commonly occurring space group, C2 at ~0.9 %. For example, if the total number of structures in a subsampling was 20,000, as with the Top10-2000 search, then $P2_1/c$ would sample 8,169 structures, while C2 would only sample 190, once the top 10 space group occurrences have been normalised. As the total number of structures was further reduced to become more affordable, such as with 9,000 total structures sampled, as with Sampling A, it would be 3,676 and 86 respectively. For certain space groups this low level of sampling is likely too low to locate the minimum structure for that specific space group, as such the minimum value of 500 is used. [2]

Therefore, to avoid very low sampling in some space groups, the Sampling-A to Sampling F schemes were constructed using the same sampling targets of 2000, 1000 and 500 structures per space group, but assigned based on observed frequencies in the CSD.

[1] Case, D. H., Campbell, J. E., Bygrave, P. J. & Day, G. M. Convergence Properties of Crystal Structure Prediction by Quasi-Random Sampling. *J. Chem. Theory Comput.* **12**, 910–924 (2016).

[2] Yang, S. & Day, G. M. Exploration and Optimization in Crystal Structure Prediction: Combining Basin Hopping with Quasi-Random Sampling. *J. Chem. Theory Comput.* **17**, 1988–1999 (2021).

The text above has been added to Supporting Note 1. Also, To draw attention to the previous work which has influenced the sampling scheme choices the following underlined comment has been added to Supplementary Table 1's caption:

“Supplementary Table 1: Tables showing the different sampling schemes investigated with the total number of structures in each scheme and the number of times the global minimum structure from a comprehensive CSP sampling was recovered in the cut down samplings. Sampling scheme choices have been chosen based upon space group occurrences from the CSD [1], as well as the previous CSP studies in Ref. [2] and [3].”

Further comments have also been added to section 2.1 (Efficient crystal structure prediction sampling) of the main manuscript.

2. The authors claim that even the cheapest SG14-500 can yield molecules with competitive predicted mobilities. But the paper only states: “Four of the top 10 molecules...”, without a quantitative result for the full picture. It would be interesting if

the authors could show the RMSE or bias between the SG14-500 and the comprehensive CSP predicted mobility, or the fitness correlation scatter plot between SG14 and the comprehensive CSP.

Thank you for your suggestion, we believe that, whilst each of the aggregated searches sampled ~3000 unique molecules over the 3 repeats, the best comparison is to the 31 molecules whose mobilities were evaluated at the comprehensive CSP level of sampling, from the top 10s of each run, including the top 10 from the Reorg-EA search.

On these top molecules we have evaluated their mobilities at each of the sampling levels and plotted the value of the landscape averaged mobility or reorganisation energy for the molecules against the comprehensive CSP landscaped averaged evaluated mobility in Figure 6 and SI Figures 11 and 12. We have added Supplementary Table 3 as requested for these 31 molecules to show the Root Mean Squared Error, Mean Signed Difference, Mean Absolute Error, Kendal rank correlation and R^2 for each of the samplings to the comprehensive level to summarise the findings.

Supplementary Table 3 – The Root Mean Squared Error (RMSE), Mean Signed Difference (MSD), Mean Absolute Error (MAE), Kendal rank correlation (τ) and R^2 score for each of the top 31 molecules whose mobility was assessed at the comprehensive CSP level against the CSP-EA CSP sampling level or reorganisation energy.

Search Setting	RMSE ($\text{cm}^2/\text{V s}$)	MSD ($\text{cm}^2/\text{V s}$)	MAE ($\text{cm}^2/\text{V s}$)	τ	R^2
Reorg-EA	-	-	-	0.35	-
SG14-500	2.6	-0.53	1.86	0.52	0.39
Top5-500	1.8	0.12	1.09	0.75	0.71
Top10-500	1.75	0.12	0.97	0.74	0.73
SamplingA	0.6	0.2	0.3	0.91	0.97

Additionally, the figures have been updated to add clarity to the comparison using colours to better show which were found during the actual search for which the comparison evaluation level is being shown. Furthermore, the following text has been amended to add clarity to the comparison between the SG14-500 and Sampling A searches top molecules:

“It is particularly encouraging that CSP-EA using the computationally least expensive SG14-500 sampling returns a set of top 10 molecules whose landscape averaged electron mobilities, after comprehensive CSP, are only slightly lower than the top molecules returned from CSP-EA using the more complete Sampling A scheme (Fig. 6a). Especially considering the strong correlation Sampling A CSP evaluations show to

the more comprehensive CSP (Fig.6d) compared to its own notably weaker correlation (Fig.6c), this result demonstrates that even an incomplete CSP landscape contains sufficient information on molecules' packing preferences to guide an EA towards high performing molecules. In fact, five of the top 10 molecules (M4, M5, M7, M12 and M14) from CSP-EA searches using the most complete CSP sampling, Sampling A, were located in the top 10 from the much cheaper CSP-EA runs that used SG14-500 sampling. However, other top 10 molecules discovered from the Sampling A CSP-EAs are ranked significantly worse when evaluated at the SG14-500 level of sampling, with M2 and M21 particularly ranked poorly compared to their Sampling A and comprehensive CSP landscape averages. These discrepancies are due to incomplete CSP sampling with SG14-500, which risks missing low energy crystal structures with good properties. As an example, for M2, which is the top ranked molecule from the Sampling A CSP-EAs, the global energy minimum ranked crystal structure from SG14-500 CSP sampling, which adopts a γ packing structure, is ranked 7th on the comprehensive CSP landscape, where the low energy structures predominantly adopt β packing (Fig. 7).

The impact of the structures missed by the SG14-500 sampling scheme is shown in Fig.6c, which compares landscape-averaged electron mobilities from SG14-500 CSP sampling and after the comprehensive CSP sampling; the differences occur due to some of the lowest energy worse performing crystal structures not being sampled by the less complete search. The correlation, albeit weak, between landscape-averaged electron mobilities from SG14-500 and the comprehensive CSP, as well as its reduced algorithmic cost as part of an EA, highlights its potential for multi-fidelity or coarse initial searching of chemical space at reduced computational expense.”

3. The CSP-EA frameworks show good results when applied to azapentacenes. However, it is a rigid and planar structure. What about the performance for soft, flexible structures? Or salts or molecules with H-bond interactions?

The CSP workflows used in CSP-EA have been applied to study hydrogen bonded systems, as well as recent work on salts.[1-3] One of the main challenges in the field of crystal structure prediction involves the increased computational cost as the degrees of freedom in the system increase, such as those introduced by including multiple components to the crystal, i.e. $Z' > 1$, and molecular flexibility.[4] This work purposely chose the azapentacenes as an initial demonstration of the approach due to their rigid and planar structures and are now looking to investigate more complex systems with the CSP-EA approach.

We do, however, note in the conclusion that the approach of using a generational EA allows scale up with greater availability of resources due to the parallel nature of assessing each molecule in the population. In previous works, outside of an EA,

flexible-molecule CSP has been performed by treating conformers individually and performing our CSP workflow on each.[5,6] Therefore, a similar approach can be taken for molecules with limited flexibility e.g. if a molecule exhibits four conformers, then the cost of CSP in the EA can be considered to be approximately four times the cost of just treating one conformer.

- [1] Pulido, A. *et al.* Functional materials discovery using energy–structure–function maps. *Nature* **543**, 657–664 (2017).
- [2] Zhu, Q. *et al.* Analogy Powered by Prediction and Structural Invariants: Computationally Led Discovery of a Mesoporous Hydrogen-Bonded Organic Cage Crystal. *J. Am. Chem. Soc.* **144**, 9893–9901 (2022).
- [3] O’Shaughnessy, M. *et al.* Porous isorecticular non-metal organic frameworks. *Nature*, **630**, 102–108 (2024).
- [4] Beran, G. J. O. Frontiers of molecular crystal structure prediction for pharmaceuticals and functional organic materials. *Chem. Sci.* **14**, 13290–13312 (2023).
- [5] Ward, M. R. *et al.* Pushing Technique Boundaries to Probe Conformational Polymorphism. *Crystal Growth & Design* **23**, 7217–7230 (2023).
- [6] Cui, P. *et al.* Mining predicted crystal structure landscapes with high throughput crystallisation: old molecules, new insights. *Chem. Sci.* **10**, 9988–9997 (2019).

Into section 2.2 we have added the following underlined text to clarify flexibility is not being considered within this study:

“Polyaromatic compounds, such as azapentacenes, have shown high charge carrier mobilities due to their delocalised π -orbitals.[22, 28, 41–43] There are four common packing motifs typically adopted by this class of molecules: β , γ , herringbone and sandwich herringbone (Fig. 3). Packings that promote greater orbital overlap result in stronger electronic couplings between neighbouring molecules in the crystal structure and therefore increase the likelihood of such crystals exhibiting higher charge carrier mobilities. Aza-substitution of polyaromatic hydrocarbons modifies both the molecular properties and the potential intermolecular interactions that define the crystal packing and has been suggested as a route to optimising charge mobility in organic semiconductors. Such molecules do not contain any flexible functional groups or rotatable bonds, simplifying the degrees of freedom required to be sampled by crystal structure prediction calculations and therefore the computational cost.[8] Thus, we choose this as a moderately-sized chemical search space to demonstrate the effectiveness of CSP-EA, and for comparison to our previous work where we presented evolutionary searches of this same chemical space using molecular properties for the fitness function. Comparison to these earlier results lets us evaluate the benefit of incorporating CSP into the fitness evaluation of the EA.”

We have also modified a paragraph in the Discussion section from:

“Further challenges to this method involve application to chemical spaces with increased molecular size and flexibility, which in turn will impact the cost of the EA search, largely due to increased cost of CSP. However, the inherently parallel nature of the generational EA used here as well as the CSP procedure allows effective scaling with improving access to computational resources.”

to

“Beyond the demonstration on azapentacenes, the CSP-EA should be applicable to molecules with stronger, more directional intermolecular interactions such as hydrogen bonds; the same CSP methods have been applied successfully to such molecules[14, 15, 50]. Further challenges to this method involve application to chemical spaces with increased molecular flexibility and to multicomponent crystals such as cocrystals and salts, where the increased structural dimensions will impact the cost of the EA search, largely due to increased cost of CSP, which would require greater sampling to generate representative crystal structure landscapes.”

4. The authors mentioned the EA is run three times per setting, and the results are averaged. However, there is no quantification of variance across runs in terms of best fitness, diversity of the results (top 10 molecules), or convergence behaviours. It will be more informative if the authors could add standard deviation, the overlap of the top 10 molecules in three runs, and the single converged trajectory for each time for Figure S10.

To demonstrate the variance of each of the sampling schemes Supplementary Figure 10 has been changed to show the standard deviation as requested. Additional Supplementary Figures 11-15 have also been added for each of the individual CSP-EA searches, to show the trajectories of each run. Furthermore, Supplementary Table 4 has been included to show the top 10 molecules found from the aggregate of the 3 repeats for each search and how many of them were found in each of the 3 repeats. Additionally, the generation in the search at which the top molecule was found has been included.

Supplementary Table 4 – Table showing for each of the evaluated search settings the number of the aggregated top 10 molecules found for each search setting found in each run. The generation the highest fitness molecule from the 3 aggregated CSP-EA repeats was found is also shown if it was found.

Search	Run		
	1	2	3

	Number of top 10 molecules found in the run	Generation best molecule found	Number of top 10 molecules found in the run	Generation best molecule found	Number of top 10 molecules found in the run	Generation best molecule found
SG14-500	4	-	6	14th	8	7th
Top5-500	10	15th	5	13th	9	12th
Top10-500	8	9th	4	-	7	15th
SamplingA	7	9th	8	14th	7	28th
GM_Search	8	-	2	17th	3	-

We have added the following text to the manuscript (p.15): “The average electron mobility across the population increases steeply over the first 10-15 generations before levelling off (Supplementary Figures 10-15), indicating that search has converged on the best molecules. The three repeats for each level of CSP sampling are quite consistent, with good overlap in the final best 10 molecules (Supplementary Figures 11-15 and Supplementary Table 4); this consistency between repeats is better for searches using landscape-averaged properties than those based on the global minimum energy structure.”

5. It would be more informative if the authors could discuss how often each molecule was selected or how stable the rankings for the top 10 molecules are. It can support the stability and reproducibility of this approach.

As mentioned in our response to the previous comment, the requested Supplementary Table 4 has been included to show the top 10 molecules found from the aggregate of the 3 repeats for each search how many of them were found in each of the 3 repeats. Additionally, the generation in the search the top molecule was found has been included.

6. The authors used B3LYP functional and 6-311+G for optimization. Is there any benchmark for it?

The B3LYP functional and 6-311+G(d,p) are used as part of the CSP workflow for optimisation of the RDKit-generated initial molecular geometry with Psi4 and for generating the molecular charge density that is used for the electrostatic model (distributed multipoles). This optimisation is also required as part of the four-point

scheme for calculation of reorganisation energies. For simplicity and cost-saving of repeated calculation, we use the same functional and basis set in both calculations.

B3LYP/6-311G(d,p) without the diffuse function can be seen to perform well for CSP. A recent study (reference 34 in the paper) used this level of theory as part of CSP for over 1000 molecules, identifying 74 % the experimentally-observed crystal structure within 2 kJ mol⁻¹ of the global minimum energy structure on the molecule's landscape. Within 8 kJ mol⁻¹ 97.8 % of the experimental structures were found.

We have added the comment: “Apart from the inclusion of diffuse functions, this functional and basis set is the same as used in our recent validation of the CSP workflow on over 1000 molecules.” in the Methods section.

For reorganisation energy a number of studies have been performed using B3LYP and varying sizes of Pople style basis sets to calculate the property for hole transport.[1-5] As the focus was on electron transport, diffuse functions were included due to the required step of optimising the anionic geometry as part of the four-point calculation.

[1] Ma, H., Liu, N. & Huang, J.-D. A DFT Study on the Electronic Structures and Conducting Properties of Rubrene and its Derivatives in Organic Field-Effect Transistors. *Sci Rep* **7**, 331 (2017).

[2] Hutchison, G. R., Ratner, M. A. & Marks, T. J. Hopping Transport in Conductive Heterocyclic Oligomers: Reorganization Energies and Substituent Effects. *J. Am. Chem. Soc.* **127**, 2339–2350 (2005).

[3] Abarbanel, O. D. & Hutchison, G. R. Machine learning to accelerate screening for Marcus reorganization energies. *The Journal of Chemical Physics* **155**, 054106 (2021).

[4] Atahan-Evrenk, S. & Atalay, F. B. Prediction of Intramolecular Reorganization Energy Using Machine Learning. *J. Phys. Chem. A* **123**, 7855–7863 (2019).

[5] Gruhn, N. E. *et al.* The Vibrational Reorganization Energy in Pentacene: Molecular Influences on Charge Transport. *J. Am. Chem. Soc.* **124**, 7918–7919 (2002).

Minor comments:

1. In Figure 1, the molecular structures are not clear and are not consistent.
2. Overall, the quality of the figures and Tables is not satisfactory.
2. The format of references is not consistent.
3. Typos throughout the text.

Thank you for drawing our attention to these issues. Figure 1 has been re-drawn with consistent molecular structure styles. Figure 3 has been improved. All other figures have been modified, using colour where appropriate to improve the representation of

the data, and increasing font size of labels. We have checked the text for typos and checked the reference formatting for consistency.

Reviewer #2 (Remarks to the Author):

Reviewer #2 (Remarks on code availability):

They have made a very detailed introduction website (<https://mol-cspy.readthedocs.io/en/latest/>), which has very detailed instructions and examples. And the results are almost the same as expected.

Thank you for checking the code and instructions.

Reviewer #3 (Remarks to the Author):

The manuscript by Johal and Day presents an evolutionary search workflow for the discovery of functional organic molecules, based on fitness measures that are obtained from the relevant crystalline forms of the candidate molecules through an implementation of crystal structure prediction algorithms.

The article uses the case of azapentacenes as a demonstration of the proposed methodology, where the charge mobility is used as the target property to be evaluated for either the global minimum of the accompanying CSP, or for a pool of low-lying minima of the energy landscape.

Overall, the manuscript is well-written and the presentation is successful in delivering the idea, showcasing its performance, and indicating the challenges and possible areas for further work.

Various interesting discussions and insights are offered, e.g., about the correlation of molecular and materials properties as optimization target and the following suggestion about using a multi-fidelity approach; and the idea of landscape-averaged properties for the EA are in particular interesting, especially for its relevance to the cases where "the fitness evaluation has no molecular property analogues".

There are, however, a few points about the presentation of the materials that can hinder the usefulness of the work. As a result, and given the above-mentioned quality of the work, I recommend the publication of the article in nature communications with the below-mentioned points addressed.

(1) A general point about the figures:

- Fig2: Distinguishing between dark points (Top5 and Top10) in b/w figure is very hard; and larger font sizes for the in-figure texts (labels and legends) would be very helpful.
- Figs 4, 5, and 6: The font sizes for texts are fairly small.
- Generally, why all b/w plots in online article?

Thank you for your suggestions the figures have been updated throughout, using larger font and colour for data points.

(2) Figure 2 (and earlier mention in line 140) reference the node-hours for a 40-core node. Later, (e.g., Table 2), the results are presented from runs on 128-core nodes. Although I understand that the cores' performance and parallelism efficiency might be different, but it would be better to unify the presentation (e.g., in "core-hours") for clarity.

Throughout the paper core-hours have now been used. Additionally, in the relevant figures the processor types used on the HPC machines have also been included in the figure captions.

(3) Line 302: the " average mobility for crystal structure i": is this average over the same set for which the summation is performed?

As part of the mobility calculations the mobility is determined in the cartesian directions, defining the z direction as the mobility channel with the highest mobility for the crystal. The " average mobility for crystal structure i" refers to the average isotropic value of these 3 mobility channels, rather than the mobility along specific directions in the crystal. This was chosen as the aim was to target generally high performing molecules. If specific mobility planes were targeted it would require greater control on the crystallisation process for candidates to be experimentally assessed.

To clarify the meaning the following has been added:

“where μ_i is the isotropic average for the mobility of crystal structure i, representing the average of the calculated mobilities along the different potential mobility channels. The sum is then over all low energy crystal structures, i.e. all crystal structures within 7.2 kJ mol⁻¹ of the global lattice energy minimum.”

(4) A few typos:

- Line 238: "the best molecules molecules from Reorg-EA show a mixture"
- Line 296: "A an alternative approach is include ..."
- Line 303: Un-paired open parenthesis

These typos have been addressed.

(5) The labelling of global minima:

In the text there are references to the true global minimum. As a general point, I would argue that this might not be the best approach. As a particular example, in line 278, I found the

"... the subsampled CSP found the incorrect global minimum energy crystal structure, ..."

a bit unassuming and misleading at the same time! In this case, the one that was found in CSP-EA was not "incorrect"; rather, the comprehensive CSP found a better minimum. In turn, the one that was found in the comprehensive CSP is not necessarily "correct"; as another search might land to an even better structure.

I think this classification of the global minima into correct/incorrect or true/untrue is not the best presentation.

The languages referring to correct or true structures has been removed, instead the following has been included in its stead:

“However, for the remaining molecule, M8, a lower energy structure was found when re-evaluated with more comprehensive CSP, compared to the incomplete sampling performed during the GM_Search CSP-EA. As a result, M8 was ranked considerably lower upon re-evaluation, due to the more stable crystal packing having a lower calculated mobility (Supplementary Note 6).”

and

“Even the outlier, M8, for which the lowest energy crystal structure found by sub-sampled CSP was the second lowest energy crystal structure on the comprehensive CSP landscape, has a calculated mobility for the comprehensive CSP global minimum crystal structure that is higher than any of the top molecules from Reorg-EA.”

Reviewer #3 (Remarks on code availability):

I have overviewed the code and its documentation. Although I didn't run the code in the sense of reproducing the manuscript's results; but I believe the source code is well-organized and, especially, a comprehensive online documentation is provided with

details about utilizing the code and nice examples of the usage of various implemented feature (e.g., structure generation, analysis of landscape, etc).

We thank the reviewers for their helpful comments on the manuscript. Please find our point-by-point response to reviewer comments below; reviewer comments are shown in red, followed by our response in black.

REVIEWERS' COMMENTS

Reviewer #1 (Remarks to the Author):

The authors have appropriately addressed my comments/suggestions; I'm happy to recommend publication in Nat Commun.

No revisions necessary.

Reviewer #2 (Remarks to the Author):

Reviewer #2 (Remarks on code availability):

The code is well organized and the test results are close to the data they provide.

No revisions necessary.

Reviewer #3 (Remarks to the Author):

The points mentioned in the original review are properly addressed in the revised manuscript; hence, I can recommend it for publication in the nature communications.

I just have a minor comment about the revised version: while the figures are improved significantly, I found the Figure 3 (especially sub-fig [3a]) to be replaced with a new one which is somewhat less clear compared to the original figure. I'm not sure why this change was made, however, I think the first figure (as it was) was better in showing the details.

We have revised Figure 3.